# Pyramiding of *gn1a*, *gs3*, and *ipa1* Exhibits Complementary and Additive Effects on Rice Yield

**DOI:** 10.3390/ijms232012478

**Published:** 2022-10-18

**Authors:** Meiru Li, Xiaoping Pan, Hongqing Li

**Affiliations:** 1Key Laboratory of South China Agricultural Plant Molecular Analysis and Genetic Improvement, Guangdong Provincial Key Laboratory of Applied Botany, South China Botanical Garden, Chinese Academy of Sciences, Guangzhou 510650, China; 2Guangdong Provincial Key Lab of Biotechnology for Plant Development, South China Normal University, Guangzhou 510631, China

**Keywords:** *Oryza sativa* L., gene pyramiding, *gn1a*, *gs3*, *ipa1*, yield-related traits

## Abstract

Pyramiding of quantitative trait loci (QTLs) is a powerful approach in breeding super-high-yield varieties. However, the performance of QTLs in improving rice yield varies with specific genetic backgrounds. In a previous study, we employed the CRISPR/Cas9 system to target three yield-related genes, *gn1a*, *gs3*, and *ipa1* in japonica ‘Zhonghua 11’, mutants of which featured large panicle, big grain, few sterile tillers, and thicker culm, respectively. In this paper, four pyramided lines, including *gn1a-gs3*, *gn1a-ipa1*, *gs3-ipa1*, and *gn1a-gs3-ipa1*, were further generated by conventional cross-breeding to be tested. Agronomic traits analysis showed that: (1) the stacking lines carried large panicles with an increased spikelet number in the main panicle or panicle; (2) the grain weight of the stacking lines, especially *gs3-ipa1* and *gn1a-gs3-ipa1,* were heavier than those in single mutants; (3) both *gn1a-gs3* and *gs3-ipa1* produced more grain yield per plant than single mutant lines; (4) pyramided lines were higher than single mutants and transcriptome analysis found improved expression levels of genes related to lipid, amino acid, and carbohydrate transport and metabolism in lines pyramiding three mutant alleles, possibly as a result of complementary and additive effects. Accordingly, the alteration of gene-expression patterns relating to hormone signaling, plant growth, and seed size control was characterized in pyramided lines. The present study not only investigates the effects of pyramiding genes, but also may provide an efficient strategy for breeding super-high-yield rice by reducing the time cost of developing pyramided lines.

## 1. Introduction

Rice (*Oryza sativa* L.) is one of the major staple crops around the world. As the population continues to rise and less arable land becomes available in agricultural production, improvement of agricultural productivity is in an upsurge to satisfy global food demand [1,2]. Four main component traits contributing to rice grain yield per plant are the number of panicles per plant, the number of spikelet per panicle, spikelet fertility, and grain weight [3]. Previous studies have highlighted multiple trait-related QTLs, and some of them have been functionally characterized in different rice varieties. *IDEAL PLANT ARCHITECTURE1* (*IPA1*)/*WEALTHY FARMER’S PANICLE* (*WFP*)/*SQUAMOSA PROMOTER-BINDING PROTEIN LIKE 16* (*OsSPL 14*), containing a mutation in miR156 cleavage site was found in both Taichung Native 1 (indica cultivar) and Aikawa and Shaoniejing (japonica cultivar), which were high-yielding varieties characterized with ideal plant architecture with a small number of unproductive tillers, more grains per panicle, and thick and sturdy stems [4,5]. The other yield-related allele, *Grain number 1a* (*gn1a*) encodes a CK oxidase/dehydrogenase (CKX) that catalyzes active cytokinin (CK) degradation. A mutation or reduced expression of *gn1a* causes the accumulation of CK in inflorescence meristems and an increased number of reproductive organs, thus, promoting grain production in Habataki (indica cultivar) [6]. *Gn1a* was also detected in the restorer line R1128, a parental line of hybrid rice featured giant panicles [7]. Both *DENSE AND ERECT PANICLE (DEP1)* and *Grain Size 3* (*gs3*) encode atypical G-protein γ subunits. The mutation of the *DEP1* allele in high-yield varieties of Shennong 265 and Jiahua 1 (japonica cultivar) resulted in dense and erect panicles [8], while in Minghui 63, the loss of the function of *gs3* (indica cultivar) produced long grains and an increase in grain weight [9]. There are other genes, such as *DROUGHT AND SALT TOLERANCE* (*DST*), *grain number, plant height, and heading date7* (*ghd7*) involved in regulating rice grain number [10,11]. Overall, these valuable genes provide candidates for improving yield-related traits.

Over the past decades, several rice breeding strategies contributing to the sustainable increase in rice yield have been successfully practiced. Breeding methods, such as conventional hybridization, heterosis breeding, and wide hybridization, are effective in combining yield-promoting traits from different cultivars to breed high-yielding varieties [2,12]. Wang et al. [12] noted that pyramided lines combining *QHD8* and *gs3* in Zhenshan97 from the donor parent 93–11 produced longer grains and a 53% higher yield per plant. Shuhui498 (R498, indica), a typical heavy panicle restorer line, was pyramided with *gn1a* and *gs3* alleles to produce a heavy panicle with an increased grain number and large grains [12]. Super-hybrid-line R1128 incorporates multiple elite genes, including *Sd1* for plant height, *Hd1* and *Ehd1* for heading date, *gn1a* for spikelet number, and *ipa1* for ideal plant architecture [13]. In another study, Shen et al. [14] investigated the effects of *gs3* and *gs3gn1a* in five different japonica varieties and found that *gs3gn1a* mutants were not only characterized with increased grain length, as in *gs3,* but also with a larger number of grains on main panicles. *IPA1* regulates the crosstalk between growth and defense to have impacts on both disease resistance and grain yield [15]. Miura et al. [5] reported that the pyramided line based on japonica variety Nipponbare harboring *gn1a-type 3* and *OsSPL14* produced higher grain yield than two single-gene introgressed lines, respectively. A recent report from Reyes et al. [16] indicated that *gn1a* and *ipa1* introgression lines developed by marker-assisted backcross breeding resulted in significantly higher spikelet numbers per panicle and primary branching compared with recurrent parents in selected NERICA cultivars. The modification of *gs3* or the introgressed segment carrying the *gs3* allele was found to promote rice grain length and weight [17,18,19,20], while *ipa1* led to increased grain weight and number per panicle [4,5,17]. These studies demonstrated the potential of combining genes to improve grain traits.

The key goal of crop breeding is to incorporate multiple desirable traits, such as high yield, superior quality, pest resistance, and environmental stress tolerance, into a single variety [2,17,21,22]. It is feasible to generate varieties containing multiple elite genes responsible for desirable traits; however, genes’ functioning tends to vary under different genetic backgrounds. For instance, *gn1a^Habataki^* (type 3) was observed to significantly increase grain number per panicle (GNPP) in japonica cultivars Koshihikari, Sasanishiki, and Kongyu131 [6,23]. However, in some indica cultivars, including Habataki, ST12, and ST6, according to Kim et al. [24], the *gn1a-type 3* allele was not found to be effective in promoting GNPP. Since QTLs could have opposing effects on grain yield [14], the genetic effect of candidate genes needs to be clarified before pyramiding them under new genetic background.

In our previous study, we successfully edited four yield-related genes (*Gn1a*, *DEP1*, *GS3*, and *IPA1*) using the CRISPR/Cas9 system in “Zhonghua11” and obtained phenotypes similar to those previously described [17]. In the current work, we designed four combinations of *gn1a-gs3*, *gn1a-ipa1*, *gs3-ipa1*, and *gn1a-gs3-ipa1* by conventional crossing. The objectives of this study include: (1) to stack multiple single elite genes contributing to a large panicle and big grain into japonica ‘Zhonghua 11’, and to produce a high-yielding plant with thicker culm; (2) to study the interplay of *gn1a*, *gs3*, and *ipa1* in regulating rice yield-related traits; (3) to generate a series of rice core germplasms with different desirable traits, and to provide a theoretical basis for molecular-designed breeding of super-high-yield japonica rice cultivars in tropical area.

## 2. Results

### 2.1. Grain Yield of gn1a-gs3, gn1a-ipa1, gs3-ipa1, and gn1a-gs3-ipa1

#### 2.1.1. Gn1a-Gs3

Grain number and grain weight are two important traits contributing to rice grain yield, and the simultaneous improvement of grain size and number is a challenge in rice production. Since larger grain often comes with a decreased grain number of grains [18,19]. Among the genes responsible for grain number and grain size, *gn1a* is considered to be one of the determinant factors that can produce heavy panicles by increasing grain number [6,10,25], while *gs3* led to increased grain size and weight [9,15,17,18,19,22]. It is, thus, interesting to explore the co-effect of two genes on grain traits. Our study pyramided null *gn1a* and *gs3* alleles into the ‘Zhonghua 11’ background to produce *gn1a-gs3* and investigate their effects on grain size and grain number. As it was shown in Figure 1, *gn1a* produced, on average, 66.1 more spikelets per the main panicle or 35.6 per panicle when compared with WT. Accordingly, the primary branching and length of the main panicle in *gn1a-gs3* increased by an average of 1.5 and 2.72 cm, respectively, when compared with WT. Compared with single *gs3*, *gs3* pyramided *gn1a* increased the spikelet number per the main panicle or per panicle, the primary branching number, and the main panicle length; there was no significant difference among these traits between *gn1a* and *gn1a-gs3*, suggesting the dominant role of *gn1a* in controlling panicle architecture and spikelet number.

The grain in *gs3* was longer and heavier compared with WT, whereas no significant difference was observed between the *gn1a* grain and WT grain (Figure 2). The 1000-grain weight of *gn1a-gs3* maintained the same level as that of *gs3*; however, the value rises compared with *gn1a* or WT, indicating the dominant role of *gs3* in regulating grain size and weight. Overall, WT, *gn1a*, *gs3*, and *gn1a-gs3* had no significant difference in seed setting ratio and effective tiller number, and *gn1a-gs3* produced the highest grain yield per plant at 31.03g/plant on average among *gn1a*, *gs3*, and *gn1a-gs3* (Figure 2). In our study, the combination of *gn1a* and *gs3* resulted in a superimposed effect on grain number and weight and was responsible for the heavy panicle phenotype. This strategy of pyramiding *gn1a* and *gs3,* thus, shows its potential for simultaneous improvement of grain size and grain number in ‘Zhonghua 11’.

#### 2.1.2. gn1a-ipa1

There was no significant difference in the length of the main panicle between WT and *ipa1*; whereas, *ipa1* produced on average 197.5 spikelets in the main panicle, less than that in *gn1a* (209.65). However, when counting the spikelet number per panicle level, *ipa1* (163.75) is significantly more than that in *gn1a* (144.85) ( Figure 1), these results suggest that *gn1a* has a strong effect on increasing the number of spikelets on the main panicle. After pyramiding *ipa1* with *gn1a*, the length of the main panicle increased to become comparable to *gn1a* at an average of 23.69 cm*;* however, *gn1a-ipa1* produced more spikelet per panicle (172.85 ) and more primary branching (17.05) than *gn1a* (144.85, 13.85), and more spikelet per the main panicle (240.3) than *gn1a* (209.65) (Figure 1), suggesting the dominant role of *gn1a* in controlling panicle size and *ipa1*, in promoting panicle branching, respectively. The combination of *gn1a* and *ipa1* resulted in a superimposed effect on the increasing spikelet number in japonica ‘Zhonghua 11’ (Figure 1).

The 1000-grain weight of *ipa1* was heavier (25.60g) compared with WT (23.27g) and *gn1a* (23.20g) (Figure 2). The 1000-grain weight of *gn1a-ipa1* (25.72g) was more than those of *gn1a* and *WT*, but close to *ipa1* grain (Figure 2). Overall, *gn1a*-*ipa1* showed its potential for the simultaneous improvement of spikelet number and grain weight.

In comparison to WT and the *gn1a*, seed setting rates in *ipa1* and *gn1a-ipa1* dropped by about 10%, suggesting the low-seed setting rate of *gn1a-ipa1* was caused by *ipa1*. The grain yield of *gn1a-ipa1* fell between those of *gn1a* and *ipa1* (Figure 2).

#### 2.1.3. gs3-ipa1

In the current study, as it was shown in Figure 1, the number of spikelets per panicle in *gs3-ipa1* had no significant difference to those in *ipa1*, whereas *gs3-ipa1* on average produced significantly more spikelets per the main panicle (240.85) and longer main panicle (24.92 cm) than in the single mutants *gs3* (153.95, 22.53 cm) and *ipa1* (197.5, 22.46 cm). The results demonstrated the superimposed effect of *gs3* and *ipa1* on increasing the spikelet number and panicle size, and the dominant role of *ipa1* in controlling the number of primary branching and spikelet numbers per plant in *gs3-ipa1*.

There is no significant difference in grain size, 1000-grain weight, and seed setting ratio between *gs3-ipa1* and *gs3*; however, the values of 1000-grain weight increased when comparing *gs3-ipa1* (27.69 g) with the single mutant line *ipa1* (25.60 g) (Figure 2). The results showed the critical role of *gs3* in regulating grain weight in *gs3-ipa1*.

The grain yield of *gs3-ipa1* significantly increased (32.63g/plant) compared with single mutant *gs3* (24.31 g/plant ) and *ipa1* (16.10 g/plant) (Figure 2). It was prompted that pyramiding *gs3* and *ipa1* resulted in a superimposed effect in promoting grain number, grain weight, and panicle size under the ‘Zhonghua11’ genetic background. The above results showed that pyramiding *gs3* and *ipa1* has great potential in breeding super-high-yield rice.

#### 2.1.4. gn1a-gs3-ipa1

By stacking *gn1a*, *gs3*, and *ipa1* together, we obtained the homozygous line of *gn1a-gs3-ipa1*. The main panicle length of *gn1a-gs3-ipa1* (25.5 cm) was significantly longer than in *gn1a-gs3* and *gn1a-ipa1*, indicating a larger superimposed effect on the length of the main panicle. The number of the primary branching of *gn1a-gs3-ipa1* significantly increased by 34%, compared with *gn1a-gs3*, but was not significantly different to that of *ipa1*, *gn1a-ipa1*, and *gs3-ipa1* (Figure 1). This result supported the hypothesis that two mutations *gn1a* and *gs3* did not impact the increased primary branching number caused by *ipa1* in *gn1a-gs3-ipa1*. There was no significant difference in spikelet number per the main panicle or per panicle among *gn1a-gs3*, *gn1a-ipa1*, *gs3-ipa1*, and *gn1a-gs3-ipa1* (Figure 1).

The lines pyramiding *gs3*, such as *gn1a-gs3*, *gs3-ipa1*, and *gn1a-gs3-ipa1* all featured the long grain phenotype as in *gs3*, indicating the key role of *gs3* in regulating grain size among the lines. The comparison of 1000-grain weight were listed as following: *gn1a-gs3-ipa1* > *gs3-ipa1* > *gs3-gn1a* = *gs3* > gn1a-ipa1 = *ipa1*, which suggested the synergistic interplay of *gs3*, *ipa1*, and *gn1a* in promoting grain weight (Figure 2). The seed setting ratio of *gn1a-gs3-ipa1* was 78.75%, which is close to *gn1a-ipa1* (76.27%) and *ipa1* (78.23%), but lower than that of *gn1a-gs3* (85.34%) and *gs3-ipa1* (88.46%), indicating the dominant role of *ipa1* in decreasing seed setting rate. Overall, there was a greater improvement of grain yield per plant of *gn1a-gs3-ipa1* (26.12 g/plant) than that of *gn1a-ipa1* (22.40 g/plant*)*, but lower than that of *gn1a-gs3* (31.03 g/plant) and *gs3-ipa1* (32.63 g/plant) (Figure 2).

### 2.2. Plant Architectures of gn1a-gs3, gn1a-ipa1, gs3-ipa1, and gn1a-gs3-ipa1

As it was shown in Figure 3, *gs3*, *gn1a*, and *ipa1* plants were taller than WT. After *gs3* was stacked to *gn1a* and *ipa1,* respectively, both *gn1a-gs3* (99.57 cm) and *gs3-ipa1* (112.84 cm) were on average much higher than single mutant *gs3* (91.65 cm), *gn1a* (93.9 cm), and *ipa1* (101.62 cm). The *gn1a-ipa1* plant was higher than *gn1*a but not significantly different to the *ipa1* plant. Triple mutant *gn1a-gs3-ipa1* (115.59 cm) was even higher than *gn1a-gs3*, *gn1a-ipa1,* and *gs3-ipa1* (112.84 cm). These results indicated that *gs3*, *gn1a,* and *ipa1* might have a superimposed effect on increasing plant height, while *gn1a* might only promote *gs3’*s height instead of *gs3-ipa1’*s.

The *ipa1* plant was characterized by fewer sterile tillers (three to five tillers per plant) than WT, while it had a minor impact on the number of effective tillers in *gn1a* and *gs3* (Figure 3). Pyramided lines containing *ipa1*, such as *gn1a-ipa1* and *gn1a-gs3-ipa1*, had more effective tillers compared with *ipa1*, but less than *gn1a* and *gs3*. Since there is no significant difference in effective tiller number between *gs3-ipa1* and *gs3*, ‘the fewer effective tiller number’ conferred by ‘*ipa1*’ might be mitigated by either *gn1a* or *gs3*. Overall, *gn1a-ipa1* and *gn1a-gs3-ipa1* produced less grain yield per plant (22.39 g, 26.12 g) compared with *gn1a-gs3* (31.03 g) and *gs3-ipa1 (32.63 g)* due to less effective tiller and seed setting rate, possibly caused by the ‘*ipa1* effect’ (Figure 2 and Figure 3). All pyramided lines containing *ipa1*, such as *gs3-ipa1*, *gn1a-ipa1*, and *gn1a-gs3-ipa1* came up with thicker culm, similar to *ipa1*, suggesting the dominant role of *ipa1* in regulating culm strength in *gn1a* and *gs3’s* genetic background.

### 2.3. Gene Networks Related to Different Agronomic Traits

To unravel the underlying mechanisms of agronomic trait formation in different pyramided lines, we performed transcriptome analysis of the materials using a young panicle (Appendix A). A comparison of the DEGs (differentially expressed genes) revealed that pyramided lines had overlapping DEGs with single mutant lines to different extents. When overlapping DEGs between *gn1a-gs3* and *gn1a*, the *gs3* levels are 6.9% and 6%, respectively, and the DEGs that exclusively exist in *gn1a-gs3* are 43.4%. In the *gn1a-ipa*1 line, overlapping DEGs between *gn1a-ipa1* and *gn1a*, the *ipa1* levels are 4.6% and 14%, respectively, and the DEGs that exclusively existed in *gn1a-ipa1* were 23%. In the *gs3-ipa1* line, overlapping DEGs between *gs3-ipa1* and *gs3*, the *ipa1* levels were 5.3% and 14.6%, respectively. A total of 20.7% DEGs was found to exclusively exist in *gs3-ipa1*. In the *gn1a-gs3-ipa1* line, overlapping DEGs between *gn1a-gs3-ipa1* and *gn1a*, the levels of *gs3* and *ipa1* were 5.4%, 5%, and 19.5%, respectively, and 21.3% of DEGs exclusively existed in *gn1a-gs3-ipa1* (Appendix A). The above results suggested that *ipa1* had a dominant function in pyramided lines, similar to the results of the agronomic traits (Figure 1, Figure 2 and Figure 3).

Next, the DEGs were classified with COG (Clusters of Orthologous Groups of proteins). Ten items based on their function correlated with energy and yield were selected (Figure 4). Compared with both *gn1a* and *gs3*, more DEGs in *gn1a-gs3* existed in the categories “Cell cycle control, Nucleotide transport and metabolism, and Cell motility”, suggesting that cell division and survival in *gn1a-gs3* were significantly influenced. This effect might be attributed to the pyramided *gn1a*, which stimulates cytokinin signaling. It is noted that DEGs in transcription and energy production and conversion were enriched in *gn1a* and in *gs3*, respectively, while *gn1a-gs3* was between that of *gn1a* and *gs3* (Figure 4A). It strongly supported the hypothesis that the pyramiding of *gn1a* and *gs3* had additive effects on the increased yield (Figure 1 and Figure 2). In *gs3-ipa1* (Figure 4B), DEGs focused on the items “Amino acid transport and metabolism, Carbohydrate transport and metabolism, lipid transport and metabolism”, which indicated that the combination of *gs3* and *ipa1* could enhance biomass production to be correlated well with the increased plant height and grain yield in *gs3-ipa1* (Figure 1, Figure 2 and Figure 3). We also found that *ipa1* had profound effects on cell cycle control and signal transduction. In *gn1a-ipa1* (Figure 4C), more DEGs enriched in the items “energy production and conversion, Amino acid transport and metabolism, and carbohydrate production and metabolism”, suggesting the yield potential of *gn1a-ipa1* double mutants by simultaneously increasing the spikelet number and grain weight (Figure 1 and Figure 2). Finally, in *gn1a-gs3-ipa1* (Figure 4D), we found DEGs enriched in the items “Amino acid transport and metabolism, and carbohydrate transport”, suggesting the additive effect of the triple mutants on yield potential (Figure 1). That is, *gn1a* had profound effects in regulating transcription, cell wall/membrane/envelope biogenesis, and signal transduction; *gs3* had major effects on energy conversion and amino acid transport and metabolism, whereas *ipa1* might function in cell cycle and cell motility, resulting in the increased spikelet number, plant height, and 1000-grain weight of *gn1a*-*gs3*-*ipa1* (Figure 1, Figure 2 and Figure 3).

Hormones play important roles in controlling plant growth and development and pyramided lines were found to alter the expression pattern of many hormone-related genes (Appendix A). According to transcriptome data, we found at least 24 auxin-related genes to be differentially regulated. These genes were associated with indole-3-acetic acid-amido synthetase, auxin-induced SAUR-like protein, auxin-responsive protein, auxin response factor, auxin efflux carrier, auxin transport, and others. Ten cytokinin-related genes belong to cytokinin-O-glucosyltransferase, tRNA isopentenyltransferase, cytokinin dehydrogenase, cytokinin riboside 5′-monophosphate phosphoribohydrolase, and others, respectively. In addition, three gibberellin-related genes, five brassinosteroid-related genes and one strigolactone-related gene were also identified. Comparison of gene expression in different lines showed that many of the DEGs in single mutant did not appear in a double mutant, while the triple mutant had more overlapping DEGs with double mutants. These results suggested that complementary effects might happen when pyramiding the three mutations.

Finally, we examined transcriptome data from young panicle materials (Appendix A) and found altered transcription levels of several genes related to plant height and grain size. The expression pattern of these genes was further confirmed by qRT-PCR.

Transcription factors belonging to the bHLH group are important in regulating rice leaf inclination and grain size. We found two bHLH group proteins Os02g0747900 (*BRASSINOSTEROID UPREGULATED 1*-*LIKE1;* *OsBUL1*/*POSITIVE REGULATOR OF GRAIN LENGTH 2*; *PGL2*) and Os09g0510500 were upregulated in *gs3-ipa1* rather than in a single mutant (Figure 5). Os02g0747900 is preferentially expressed in the lamina joint controlling cell elongation and affecting leaf angles. Os02g0747900 knockout mutant (*osbul1*) produced erect leaves and smaller grains, whereas overexpression of *OsBUL1* increased lamina inclination and grain size [26]. Os09g0510500 is another bHLH transcriptional activator which interacts with LO9-177 and forms a trimer complex with LO9-177 and OsBUL1, promoting laminar joint cell elongation and influencing plant height and grain size. The above two proteins acted in the same complex to regulate leaf inclination and grain size, suggesting that upregulation of the two genes’ expression played a positive role in the increased grain size and plant height in *gs3-ipa1* (Figure 2 and Figure 3).

Os03g0179400 *(GROWTH UNDER DROUGHT KINASE; GUDK*) is a drought-inducible receptor-like cytoplasmic kinase and secures grain yield under drought and well-watered conditions. The *gudk* mutants feature a significant grain-yield reduction under normal well-watered conditions or when facing drought stress at the reproductive stage [27]. We observed this gene was down-regulated in *gs3-gn1a*, *gs3-ipa1* and *gn1a-gs3-ipa1*(Figure 5), however, without significant grain yield reduction (Figure 2), which suggested the penalty on yield loss might be mitigated by other factors.

Os05g0156900 (*Chalk5*) influences grain chalkiness, rice yield, and many other quality traits. It encodes a vacuolar H^+^-translocating pyrophosphatase (V-PPase) with inorganic pyrophosphate (PP_i_) hydrolysis and H^+^-translocation activity. Elevated expression of *Chalk5* increases the chalkiness of endosperm and produced chalky grain [28]. This gene was downregulated in *gs3*, *gn1a-gs3,* and *gn1a-gs3-ipa1* (Figure 5), suggesting *gs3’*s potential in decreasing the chalkiness and improving grain quality. Os06g0650300 (*OsglHAT1*) encodes a new-type GNAT-like protein with intrinsic histone acetyltransferase activity. Elevated *OsglHAT1* expression enhances grain weight and yield by enlarging spikelet hulls via increasing cell number and accelerating grain filling, and increases global acetylation levels of histone H4 [29]. We detected the increased expression of this gene in *gn1a-gs3*, *gn1a-ipa1,* and *gn1a-gs3-ipa1*(Figure 5), which might contribute to the big grain phenotype (Figure 2). Os01g0177400 (*dwarf 18*) encodes a gibberellins 3β-hydroxylase, and its knockout mutant exhibits extremely dwarf phenotypes suitable for large-scale indoor planting [30]. This gene was upregulated in *ipa1*, *gn1a-gs3,* and *gn1a-gs3-ipa1* (Figure 5) and conferred their tall statures (Figure 3).

Taken together, the above results suggested that different plant height and grain size-related genes were regulated in pyramided lines and potentially contributed to variations in phenotypes.

## 3. Discussion

### 3.1. gs3’s Potential in Producing Big Grain Lines

Grain length is a critical trait affecting grain size, shape and appearance, and rice yield. Mutation of *gs3* resulted in a long grain and heavy grain weight, demonstrating its role to be one of the most important regulators of grain size in rice breeding [9,18,19,20,22,25,31], and CRISPR/Cas9 is a convenient approach to obtain *gs3* mutant [17]. Rice quality may be further improved by stacking other favorable alleles with *gs3*, for instance, the combination of the pair of ISA1 for low chalkiness and *gs3* considerably improved the grain quality of ZS97 [32]. In Wang et al. [13], *gs3* pyramiding with spl16 produced slender grain and an ideal appearance contributing to improved grain quality and grain yield.

*Gs3* loss-of-function allele was reported to promote grain length but negatively impact grain number in panicles, which is a result of a genetic trade-off between grain size and grain number [9]. However, in our study, the *gs3* mutant increased grain length without a negative impact on grain number (Figure 1 and Figure 2), and moreover, promoting effects of grain length, weight, and number were found in all pyramiding lines containing *gs3* including *gn1a-gs3*, *gs3-ipa1*, and *gn1a-gs3-ipa1* (Figure 1 and Figure 2). As a positive regulator of the chalkiness of endosperm and to affect grain quality, *Os05g0156900*, was downregulated in *gs3*, *gn1a-gs3*, and *gn1a-gs3-ipa1* (Figure 5). Overall, these results demonstrated *gs3*’s potential to simultaneously improve grain weight, grain number, and grain quality by pyramiding with *gn1a* and *ipa1*. Previous studies pyramiding *gn1a* and *gs3* alleles in indica rice Shuhui498 and in five different japonica varieties also found increased grain length and grain number on main panicles [12,14].

### 3.2. ipa1 as a Candidate for ‘Ideal Plant Architecture’

Tiller number and panicle size are closely related to rice yield while ‘Idea Plant Architecture’ was supposed to be a new breeding approach to encourage productivity [4,5]. IPA was represented by fewer sterile tillers, larger panicles, and stronger culms attributing to *ipa1* protein accumulation, resulting in an increased yield and lodging resistance. It has been widely identified in a number of japonica and indica varieties and, by conventional breeding and the marker-assisted selection (MAS), introgression of the *ipa1* allele showed its potential in breeding novel high-yielding lines [24]. It is convenient to apply CRISPR/Cas9 editing and to obtain the *ipa1* phenotype by targeting the region containing the OsmiR156 target site [17]. In our study, *ipa1* exhibited ‘Ideal Plant Architecture’ and featured a reduced effective tiller number, increased grain number and weight per panicle, and a strong stem (Figure 1, Figure 2 and Figure 3). The lines containing *ipa1*, such as *gn1a-ipa1*, *gs3-ipa1*, and *gn1a-gs3-ipa1,* were characterized by thick culm, similar to *ipa1* (Figure 1, Figure 2 and Figure 3). It was noted that though a reduced effective tiller number was observed in *gn1a-ipa1* and *gn1a-gs3-ipa1*, it did not occur in *gs3-ipa1*. In comparison with *gn1a*, *gs3* significantly disturbed *ipa1*’s role in regulating tillering. We observed ‘the *ipa1* effect’ among all the pyramided lines containing *ipa1,* characterized by increased primary branching number, spikelet number, and culm strength and transcriptome data analysis also confirmed *ipa1*’s dominant function in pyramided lines (Figure 4). Additionally, the mutation of *ipa1* had a superimposed effect on grain weight and plant height when combined with *gs3*. Former studies reported that the pyramided line based on japonica variety Nipponbare harboring *gn1a-type 3* and *OsSPL14* produced higher grain yield than two single-gene introgressed lines, respectively [5]. However, another study investigating the effects of the *gn1a-type 3* allele in ST12, and ST6 of Habataki found no significant additive effect in g*n1a-type 3* [24]. These results suggest that genetic background has profound effects on the function of pyramided genes.

### 3.3. gn1a in Breeding Large Main Panicle Line

Cytokinin plays important roles in regulating plant growth and development, for example, in shoot and root growth, seed development, and apical dominance [33]. As early as in Ashikari et al. [6], *gn1a* was firstly recognized to be an elite allele in breeding super-high-yield rice, mutated *gn1a* causing cytokinin accumulation in inflorescence meristems resulted in increased spikelet number in the main panicle and enhanced grain yield. A knockdown of the *OsCKX2* expression might be achieved by the introgression of *gn1a*-mutated allele [5,23,24], shRNA-mediated *OsCKX2* gene silencing [34], or the specific knockout of *OsCKX2* by CRISPR/Cas 9 [17]. CRISPR/Cas9 editing is a convenient approach to generate a *gn1a* line. In the current study, *gn1a* was identified with a significantly increased spikelet number. As a superimposed effect in single mutant lines *ipa1*, *gs3* pyramided with *gn1a* was characterized by a large and heavy panicle by a dramatically increased number of spikelets per the main panicle or per panicle and length of the main panicle. Thus, our study demonstrates the role of *gn1a* in contributing to large panicle traits on *ipa1*, and the *gs3* genetic background. As *Gn1a* encodes a CK oxidase/dehydrogenase (CKX) that catalyzes active cytokinin (CK) degradation, it has a dominant effect on stimulating meristem cell division, and thus, increases grain number, the role of this gene is in accordance with previous studies [6].

### 3.4. Complementary and Additive Effects in Gene Networks in Pyramided Lines

By comparing the transcriptome data of a young panicle from a single mutant with pyramided lines, we found exclusive effects imposed by an individual pyramided allele in the pyramided lines (Appendix A). DEGs analysis revealed that *gs3* contributed specifically to energy conversion, lipid and carbohydrate transport and metabolism, *gn1a* in transcription and signal transduction, and *ipa1* in cell cycle, cell motility, and signal transduction, which correlated well with the traits in a single mutant (Figure 4). It was found to be a dominant effect of *gn1a* on transcription and signal transduction in all of the pyramided lines, which suggested its impact on cytokinin content had predominant effects over the others. In *gs3-ipa1*, DEGs related to carbohydrate transport and metabolism were highly enriched, which had a positive relationship with its improved plant height and big grain size (Figure 2, Figure 3 and Figure 4). In *gn1a-gs3*, the level of DEGs in different items was becoming intermediate between *gn1a* and *gs3*, suggesting complementary effects of the two in *gn1a-gs3* (Figure 1, Figure 2, Figure 3 and Figure 4). In *gn1a-ipa1* and *gn1a-gs3-ipa1*, the DEGs in carbohydrate transport and metabolism were also moderately enriched, probably relating to the dynamic of growth and grain size. Altogether, the pyramiding of *gs3*, *gn1a,* and *ipa1* produced complementary and additive effects on plant growth and grain size (Figure 1, Figure 2, Figure 3 and Figure 4).

Plant hormones, auxin, cytokinin and BR play important roles in regulating plant growth and development. By analyzing the transcriptome data (Appendix A), we found that more hormone-related genes were regulated in pyramided lines than in a single mutant. For example, the expression of indole-3-acetic acid-amido synthetase GH3.2 and GH3.6 were upregulated exclusively in pyramided lines. There were also expression patterns of single-mutant genes distinctive to pyramided lines. For example, *OsARGOS* for auxin-regulated genes involved in organ size was upregulated only in *gn1a-gs3*. Other genes showed similar expression levels in single mutant and pyramided lines, for example, the expression of *OsTCP17* in *gs3* and *gs3-ipa1*. In DEGs related to cytokinin, we found four cytokinin dehydrogenases, whose expressions were downregulated in the pyramided lines but not in single mutants, suggesting that modified cytokinin content could be the major reason responsible for the variation in traits (Figure 1, Figure 2 and Figure 3).

Several DEGs related to grain size were identified analyzing transcriptome data (Appendix A). For example, *OsSGL,* which regulates stress tolerance and grain length; *Os03g0179400*, a drought-inducible receptor-like cytoplasmic kinase and regulator of grain yield; *OsglHAT1*, a histone H4 acetyltransferase regulating grain weight, yield, and plant biomass. These genes showed different expression levels in pyramided lines than in single mutants. Interestingly, three bHLH transcription factors were found in datasets and *OsBUL1* and *OsBC1* were only upregulated in *gs3-ipa1*. As the two proteins were reported to form a complex together with LO9-177 in controlling grain size and leaf inclination, up-regulation of the two genes may be responsible for the increased plant height and grain size (Figure 2, Figure 3 and Figure 5).

### 3.5. Tradeoffs among Grain Weight, Grain Number, Plant Height, and Lodging Risk

Grain weight and number, panicle number, and panicle size are major components in rice yield. However, these traits are often negatively correlated. Thus, the combination of multiple desired traits by a conventional cross-breeding method posed a daunting challenge in rice production.

*Gn1a* and *ipa1* promote grain number, and *ipa1* and *gs3* are positive regulators of grain weight. Additionally, *gn1a* contributes to long panicles and *ipa1* and *gn1a* produce more primary branching numbers. In our study, single mutant lines, *gn1a*, *gs3*, and *ipa1* were all characterized with heavy panicles in comparison to WT (Figure 1). Furthermore, the superimposed effect on both grain number and weight was characterized in *gn1a-gs3*, *gn1a-ipa1*, and *gn1a-gs3-ipa1* over their respective single mutant lines (Figure 1 and Figure 2). Thus, our data strongly supported the hypothesis that stacking *gn1a*, *gs3*, and *ipa1* together is an approach to promote both grain weight and number, to tackle the usually negative-correlated relationship between grain weight and number in rice breeding. It also demonstrates the potential of pyramiding these elite alleles in breeding high-yielding rice.

Plant height is one of the most important agronomic traits affecting rice yield. Semi-dwarfism architecture has the advantage of contributing to lodging resistance and increased tiller number, but it usually comes with short panicles which negatively impacts rice yield. On the contrary, increased plant height improves high-grain yield, but suffers from lodging because of its high stature. Lodging dramatically damages rice yield and plant height and lodging resistance are usually negatively correlated [35]. In this study, *gn1a-gs3*, *gn1a-ipa1*, *gs3-ipa1*, and *gn1a-gs3-ipa1* lines were susceptible to lodging in field trials in Guangzhou, Southern China, where rainstorms and typhoons were prevalent. Long or heavy panicles produced by these plants and their tall stature come with an increased risk of lodging. Nevertheless, simultaneously increasing both grain weight and number to boost rice-yield potential might be achieved by pyramiding *gn1a*, *gs3*, and *ipa1*. Further studies are required to investigate the performance of the above lines in different field trials before realizing their potential in the future.

In summary, the superimposed effect on grain weight and number by pyramiding *gn1a*, *gs3*, and *ipa1* showed the potential of a molecular-designed breeding method in pyramiding alleles for super-high-yield elite cultivars; additionally, the superimposed effect on the plant height by pyramiding *gn1a*, *gs3*, and *ipa1* may exhibit the lodging risk.

## 4. Materials and Methods

### 4.1. Generation of gn1a-gs3, gn1a-ipa1, gs3-ipa1, gn1a-gs3-ipa1

*Gn1a*, *gs3*, and *ipa1* mutants were previously obtained via CRISPR/Cas9. Embyogenic calli from a japonica cultivar “Zhonghua11” were used for Agrobacterium-mediated transformation with CRISPR vectors targeting *gn1a, gs3* and *ipa1.* Transgenic plants were obtained and the targeting sites were PCR amplified and sequenced [17]. T2 generation of homozygous mutant *ipa1* (with a 21-bp deletion), *gn1a* (with a 1-bp deletion), and *gs3* (with a 47-bp deletion and 23-bp insertion) were selected to produce gene-stacking lines by conventional crossing. Homozygous double mutants, *gn1a-gs3*, *gn1a-ip1a*, and *gs3-ipa1*, and triple mutant *gn1a-gs3-ipa1* were examined by PCR amplification of the target sites, further confirmed by sequencing (Appendix A).

### 4.2. Agronomic Trait Analysis

T3 generation homozygous lines (*gn1a, gs3, ipa1, gn1a-gs3*, *gn1a-ipa1*, *gs3-ipa1*, and *gn1a-gs3-ipa1*) and wild type (WT) plants, were planted in experimental fields at South China Normal University in Guangzhou, P. R. China. Their phenotypes were investigated in three successive years from 2019 to 2021. The lines were sowed on 10th March each year and harvested on 30th June. At the mature stage, major agronomic traits were recorded from 20 plants of each line, which were randomly chosen from 50 tested plants. These traits include plant height, effective tiller number, main panicle length, spikelet number per the main panicle, spikelet number per panicle, number of primary per the main panicle, 1000-grain weight, and grain yield per plant. Since results generated from each of the three years provide similar implications, only the representative data from 2021 was presented in the current study.

### 4.3. RNA-Seq and Analysis

For RNA-Seq analysis, total RNA was extracted from 10 young panicles (about 1 cm) from each line using TRIzol^®^ reagent (AmbionTM, Lot No. 15596018) according to the manufacturer’s protocol. RNA-Seq was performed by Novogene Bioinformatics Technology Co., Ltd. (Tianjin, China) with Hiseq-PE150 (Illumina, Inc. San Diego, CA USA) and the raw data was analyzed using the software package of Transcriptome Analysis with Reference Genomes in the BMKCloud cloud server2. Gene expression level quantification was estimated by fragments per kilobase of transcript per million fragments mapped (FPKM). A value of |log2(foldchange)| ≥ 1 (FDR < 0.01) was set as the threshold for differentially expressed genes (DEGs). Data from two biological replicates were used for the analysis.

### 4.4. Quantitative RT-PCR

Briefly, 10 young panicles (0.5–1 cm length) from one mutation line were collected for RNA extraction, using Trizol reagent (Invitrogen, Waltham, MA, USA). RNA samples were treated with DNaseI following the manufacturer’s protocol, and to be quantified under Nanodrop Spectrophotometer (Nanodrop Technologies, Wilmington, NC, USA). First-strand cDNA synthesis and qRT-PCR (quantitative reverse transcriptase polymerase chain reaction) were performed according to the instructions of commercial kits (TransGen Biotech, Beijing, China). PCR amplification was performed following the procedure: 30 s 94 °C preincubations, 45 cycles of 5 s denaturation at 94 °C, and 30 s annealing and extension at 55 °C. The relative expression levels were calculated by the delta-delta Ct method [36]. The Rice Actin gene (*LOC_Os03g50885*) was used as internal control and three technical replicates were designed for each sample. Primers used for qPCR are listed in Appendix A.

## Figures and Tables

**Figure 1 ijms-23-12478-f001:**
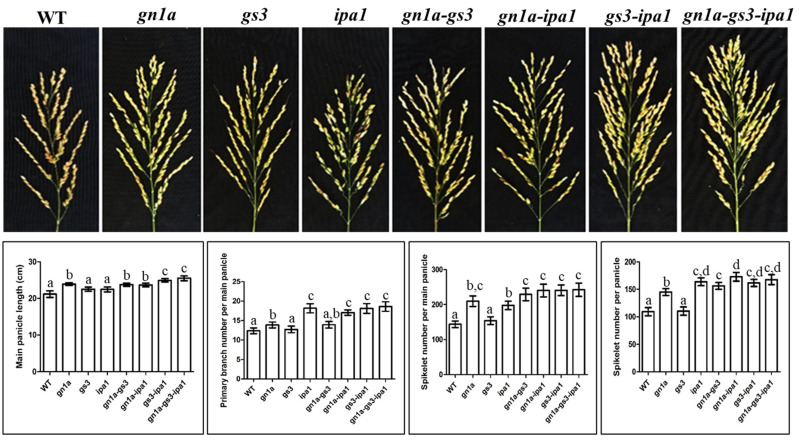
Representative main panicle architectures, main panicle length, the number of primary branching per main panicle, the number of spikelet per main panicle, the number of spikelet per panicle of the T3 generation. Means labeled with different lower letters are significantly different by the Duncan Multiple Range Test at *p* < 0.05 using the SPSS software package version 17.0 (SPSS Inc., Chicago, IL, USA).

**Figure 2 ijms-23-12478-f002:**
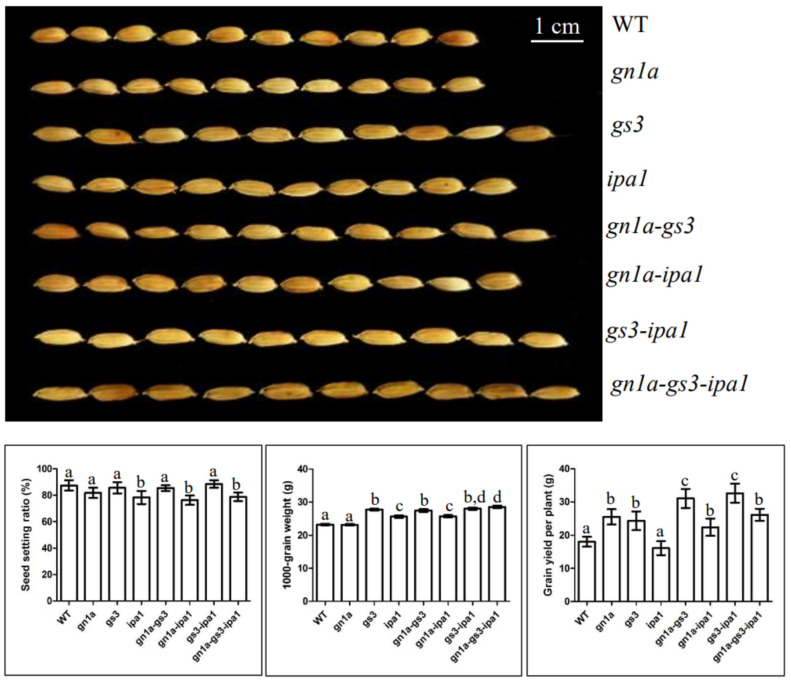
Representative grain size, seed setting rate, 1000-weight, and grain yield per plant of the T3 generation. Seed length was compared by line up of 10 seeds. Means labeled with different lower letters are significantly different by the Duncan Multiple Range Test at *p* < 0.05.

**Figure 3 ijms-23-12478-f003:**
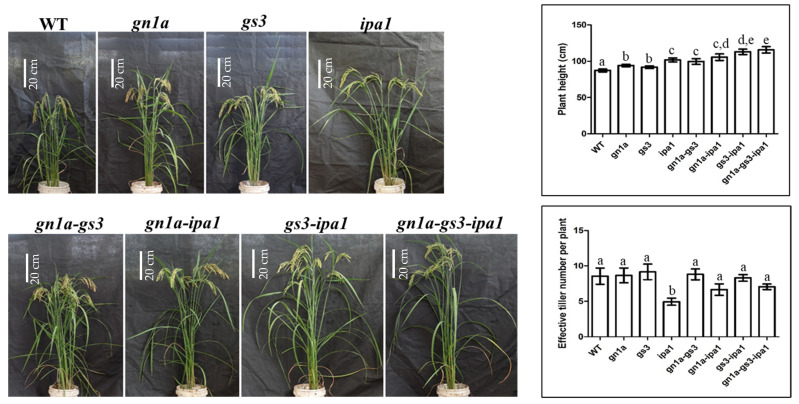
Representative plant profile, plant height, effective tiller number of the T3 generation. Means labeled with different lower letters are significantly different by the Duncan Multiple Range Test at *p* < 0.05.

**Figure 4 ijms-23-12478-f004:**
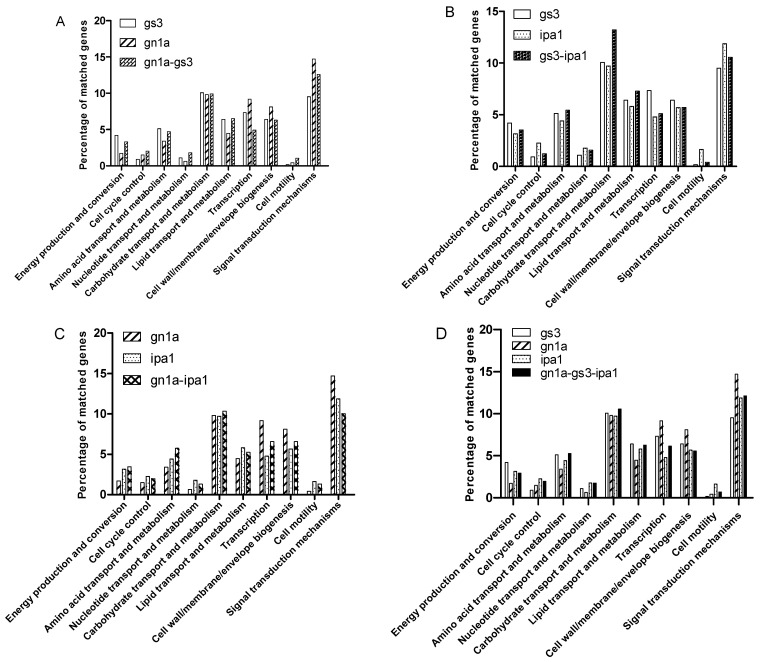
COG function classification of DEGs in different lines. (**A**) The COG function classification of DEGs in *gs3*, *gn1a* and *gs3-gn1a*; (**B**) The COG function classification of DEGs in *gs3*, *ipa1* and *gs3-ipa1*; (**C**) The COG function classification of DEGs in *gn1a*, *ipa1* and *gn1a-ipa1*; (**D**) The COG function classification of DEGs in *gs3*, *gn1a*, *ipa1* and *gn1a-gs3-ipa1*.

**Figure 5 ijms-23-12478-f005:**
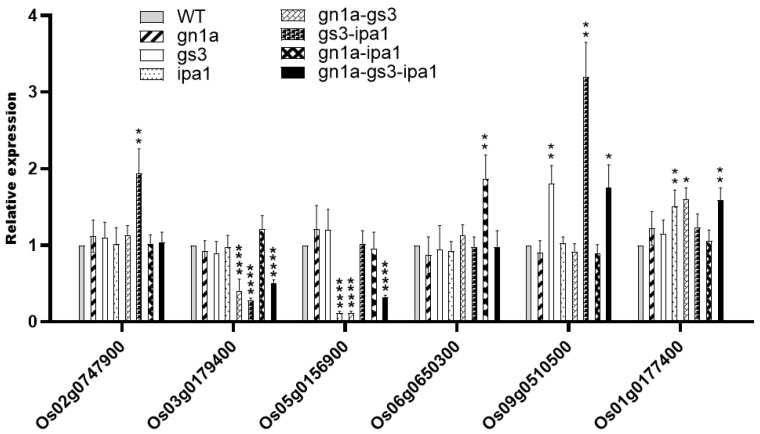
Expression of grain weight, quality, and plant height-related genes in young panicle from WT, *gn1a*, *gs3*, *ipa1*, *gn1a-gs3*, *gn1a-ipa1*, *gs3-ipa1*, *gn1a-gs3-ipa1.* Date are shown as mean ± SD from three replicates; “*”, “**” and “****” indicated significant differences at *p* < 0.05, *p* < 0.01 and *p* < 0.001, respectively.

## Data Availability

Not applicable.

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
