# Peer review of "Pyramiding of gn1a, gs3, and ipa1 Exhibits Complementary and Additive Effects on Rice Yield"

_ijms, 2022, doi:10.3390/ijms232012478_

Round 1
Reviewer 1 Report
Pyramiding of gn1a, gs3, and ipa1 exhibits complementary and 2 additive effects on rice yield by Li et al.
The manuscript describes about agronomical performance and transcriptome using the stacked line of the 3 alleles. It gave a good overview on the synergistic effect between/among gn1a, IPA1, and gs3. The experiments were soundly conducted, and explanation of the results seems almost sufficient. However, let me ask the authors to fix the problems listed below.
Major comments:
L91-L102: It doesn’t look result.
L124-136: It is not result.
L159-161: It is not results. “GS3 locus” is not correct as a term of genetics. Specify allele. Be strict for the terms “gene”, “locus” and “allele”.
Fig. 2, “Grain yield per plant”: Values are strange. This must be a critical mistake.
L181-184: It doesn’t look result.
L196, “synergistic”: For what does this mean? Choose a word to explain the relationships in a more scientific way.
Please include scale bars on your images
Minor comments:
L15, “strong culm”: It is controversy to some of the results. “Thicker” might be safer.
L18, “main panicle or panicle”: This is confusing. Clearly define what was explained.
In the introduction, the authors mentioned that one of their objectives is to produce high-yielding plants with strong culm. However, I didn’t find any experiment across the manuscript that addressed this objective.
L53, “Grains Height Date-7”: “Grain number, plant height, and heading date7”.
L105, “gn1a was recorded”: Is it correct grammatically?
L117, “similar”: This word is not appropriate to explain the results with statistical test. The regular term such as “no significant difference” or “not different significantly” should be chosen.
L138, “main panicle or panicle”: This is confusing. Fix is needed not only here but also methods, because “spikelet number per panicle” is not clearly understood.
L151, “slightly”: Avoid this word for statistic results.
L156, “slightly”: Same as above.
L161, “number of spikelets per panicle primary branching”: Can it be defined?
L169, “dramatically”: It is just an authors’ impression.
L175-176, “Over all…”: It is not a result. Moreover, do the authors need to explain about hybrid?
Fig. 2: “1000-grain weigth” looks a mistake.
L185 “slightly”: Same as above.
L187 “similar”: So different or not different? Please appreciate statistics.
L198 “close”: So different or not different? Please appreciate statistics.
L220-223, “All pyramided lines containing IPA1”: If the culms were strong, why they are susceptible to lodging (L439)?
L239 “Ten items”: Ten items of what?
Figure 4: I wonder why the authors didn’t include the background line, Zhonghua 11. If data is available, I appreciate including it.
L281-317: Use a uniform size of font. Please reconfirm all the possible references are included (It seems some references are lacked). More importantly, this part looks redundant. Specify and separate the topics for general readers.
L329, “sple16”: OsSPL16? Which allele?
L333, “However, in our study, gs3 mutant simultaneously increased grain length and weight, without the negative impact on grain number…”: Looking at figure 1, the line which carries only GS3 is not significantly different from the WT. Probably some mistake in English.
L356, “It was suggested…”: This sentence is not appropriate to put here, because this study is to find a clue to understand the “counteraction”. Why the authors return to the starting point?
L470- and L480-: Specify the numbers of biological zreplicates.
Reviewer 2 Report
The pyramding of yield related genes in different genetic backgrounds of important cereal crops such as rice, is an important development in plant breeding and must be researched. The authors have done a commendable job in this regard targeting the genes gn1a, gs3, and ipa1. However, the communication of the results to the readership needs improvement before it can be published. I have noted the following issues:
Line 85: should be 'germplasms'
- Large sections of the Results go into detail describing the function of the yield related genes and contain citations for the information. Almost all of this info should be moved to the introduction of the paper. That is where we should be told about the function of gn1a, gs3, and ipa1.
- Throughout the paper, the authors give no numerical or statistical data descriptions to the reader. Everything is described as 'higher' or 'lower'. Instead the authors should use the actual data, so the author can get an idea of how strong the statistical relationship actually is e.g. Cross #1 was 75% higher in yield compared to Cross #2 (p = <0.001).
- In Figure 2, what is the picture of various grains meant to show? There doesn't appear to be any significant difference in grain size between the images? Likewise, no mention of this image is made in the figure description.
- When looking at the transcriptome data, did you check if the key genes under examination were actually knocked out? I couldn't open the supplementary file to examine your figures.
- In your discussion, you make almost no references to other published literature to explain your results. What were the observations of other authors when pyramiding genes? What were the observations of other authors in different crops such as Maize or other cereals? Did you observe similar outcomes?
- In your Materials & Methods section, it would be good to give the reader an idea if the CRISPR Cas9 mutants were complete or partial knockouts.
